Nacre morphology and chemical composition in Atlantic winged oyster Pteria colymbus (Röding, 1798)

Santana Pablo
Aldana Aranda Dalila daldana@cinvestav.mx
Departamento de Recursos del Mar, Centro de Investigación y de Estudios Avanzados del Instituto Politécnico Nacional , Mérida, Yucatán , México
Scozzafava Andrea
Electronic publication date: 2021 Jul 15
Publication date: 2021
Volume: 9
Electronic Location ID: e11527
Received 2021 Mar 10; Accepted 2021 May 6
Copyright: © 2021 Santana and Aldana Aranda
Copyright year: 2021
Copyright holder: Santana and Aldana Aranda
License: This is an open access article distributed under the terms of the Creative Commons Attribution License, which permits unrestricted use, distribution, reproduction and adaptation in any medium and for any purpose provided that it is properly attributed. For attribution, the original author(s), title, publication source (PeerJ) and either DOI or URL of the article must be cited.
License URL: https://creativecommons.org/licenses/by/4.0/

Keywords: Nacre, Microstructure, Nanocrystals, Oyster, Chemical composition, Pteria

Funding: Consejo Nacional de Ciencia y Tecnología CVU 390072 National Laboratory of Nano- and Biomaterials (LANNBIO) CINVESTAV-IPN Unidad Merida Fomix-Yucatán and CONACyT 2008–108160 Y The research reported here was financed in part by the Consejo Nacional de Ciencia y Tecnología (Grant No. CVU 390072). X-ray diffraction and SEM analyses were done at the National Laboratory of Nano-and Biomaterials (LANNBIO), CINVESTAV-IPN Unidad Merida, and funded by the Fomix-Yucatán and CONACyT (Grant No. 2008–108160 Y). The funders had no role in study design, data collection and analysis, decision to publish, or preparation of the manuscript.

==============================
The microstructure and nanostructure of nacre in Pteria colymbus were studied with high-resolution field emission scanning electron microscopy (FESEM). The tablets were found to be flat and polyhedral with four to eight sides, and lengths ranging from 0.6 to 3.0 µm. They consisted of nanocrystals 41 nm wide, growing in the same direction. X-ray diffraction showed the crystals to be mineral phase aragonite, which was confirmed by Raman spectroscopy. Fourier transform infrared spectroscopy identified a band at 1,786.95 cm−1 attributed to carboxylate (carbonyl) groups of the proteins present in the organic matrix as well as bands characteristic of calcium carbonate. X-ray fluorescence showed the nacre to contain 98% calcium carbonate, as well as minor elements (Si, Na, S and Sr) and trace elements (Mg, P, Cu, Al, Fe, Cl, K and Zn).

Introduction

Mollusk shells are mineralized tissues that fulfill structural functions (Addadi, Raz & Weiner, 2003). In all three main mollusk classes (Cephalopoda, Gastropoda and Bivalvia) the shell consists of stratified layers, each with a unique mineral composition (Dauphin & Denis, 2000). Shell-forming crystals are organized on these layers according to different configurations which define a shell’s microstructures. Specific microstructures are characteristic of calcite (i.e., prismatic, foliate) and aragonite (i.e., nacre, laminar cross). Secreted polymorph type and microstructural types are used to characterize large mollusk groups, particularly the bivalves.

Nacre is the most studied aragonitic microstructure and is widely distributed in mollusks (Towe & Hamilton, 1967). Its stratified microstructure gives mother-of-pearl its luster and provides excellent mechanical properties. Nacre has been of great interest to the pearl industry, making it one of the most studied hard tissues (Wang et al., 2013). Because it is osteoconductive and biodegradable, interest has increased recently in technological applications related to nacre, such as the manufacture of bio-inspired super-resistant materials and clinical implants (Tang et al., 2003; Oaki & Imai, 2005).

In each of the three main mollusk classes (Gastropoda, Cephalopoda and Bivalvia) nacre exhibits specific growth patterns and mechanisms. In Gastropoda, for example, the aragonite nanocrystals in the nacre are stacked in towers and their c axes are aligned. However, they have a composite cross laminar arrangement, and a third hierarchical order of flat aragonite fibers from 50 to 100 nm thick, 300 nm wide and a few micrometers long (Romana et al., 2013). In the nautilus (Cephalopoda), nacre exhibits a mixed behavior with simultaneous growth in towers and terraces occurring in adjacent locations (Saunders & Landman, 2010). Nacre in the Bivalvia has terraced growth and the three axes of crystals are co-oriented. The order Pterioidae has shells that are unequal, monomyary, and not equilateral; the right valve is generally less convex than the left (Wada & Tëmkin, 2008; Cummings & Graf, 2015). Their shells are formed by superposition of an outer organic layer, the periostracum, a prismatic layer and the inner nacreous layer (Kennedy, Taylor & Hall, 1969).

Nacre is a biomineral consisting (by weight) of 95% aragonite (CaCO3) with the remaining 1–5% being organic matrix (Zhang & Li, 2012). Its microstructure is one of layered “brick” (aragonite tablets) and “mortar” (protein-polysaccharide matrix). This structure provides nacre with twice the strength and up to 1,000 times the toughness of its constituent components alone (Li et al., 2004; Veis & Dorvee, 2013; Morris et al., 2016). Individual nacre tablets have a nanoscale structure based on aragonite nanograins, nanoblocks and nanofibers (Wang et al., 2013). Nanoscale structural organization differs between bivalve mollusks, resulting in different tablet forms at growth completion.

Better understanding of the composition and hierarchy of biological system microstructures is key in the search for new materials and provides deeper insight into evolutionary processes (Jáuregui-Zúñiga et al., 2003; Oaki & Imai, 2005; Wang et al., 2013; Nakamura Filho et al., 2014; Wegst et al., 2015). The present study objective was to analyze the micro- and nanostructure of P. colymbus nacre with scanning electron microscopy, and its chemical composition with X-ray diffraction, X-ray fluorescence spectrometry, Fourier transform infrared spectrometry and Raman spectroscopy.

Materials & methods

Shells of Pteria colymbus were collected in the Alacranes Reef, in the state of Yucatán, Mexico. They were placed in a soap solution, cleaned, and stored at 4 °C for 48 h. To remove the inorganic and biogenic matter from shells was cleaned by ultrasound with a soap solution for 5 min (Ky, Lo & Planes, 2017). Six 1 cm2 samples were cut from the shells using a 32 mm-diameter diamond disc. Samples were washed again by ultrasound for five minutes and dried at 65 °C for 3 h (Ren et al., 2009; Xu & Zhang, 2015).

Nacre tablet morphology was characterized with a JEOL 7600F ultra high resolution field emission scanning electron microscope (FESEM). Samples were coated with Au/Pd and processed at a 1–30 kV acceleration voltage (Ren et al., 2009; Liu & Li, 2015).

For the chemical analysis, a 1 cm2 nacreous layer sample was separated from the shell with the help of a rotary tool at 2000 RPM. The sample was always kept immersed in water at room temperature. The sample was cleaned by ultrasound with a soap solution for five minutes and dried at 65 °C for three hours. Subsequently, it was placed in aluminum sample holders for analysis in the diffractometer (Bruker D8) using monochromatic CuKa radiation. The XRD patterns were collected at 20–90° (2θ) in 0.02° steps with a 0.96 s count time interval. The resulting diffraction patterns were compared with the card for aragonite (PDF no. 000411475) from the crystallographic records of the International Center for Diffraction Data (ICDD) database (Weiner & Traub, 1980; De Paula & Silveira, 2009; Heinemann et al., 2011).

To analyze the nacre by Fourier transform infrared spectrometry (FTIR) analysis, the nacre layer from the shell was separated with a rotary tool at 2000 RPM. The sample was ultrasonically cleaned with a soap solution and dried at 65 °C for 3 h. The sample was ground in an agate mortar to give the appearance of a fine powder and then dried at 65 °C for 3 h. Tablets were prepared with 0.5 mg of nacre powder and 200 mg of KBr. Infrared analyzes were run at a 4 cm resolution in two wave ranges: 400–4,000 cm−1 and 550–4,000 cm−1. The analyzes were done in reflectance mode on a FTIR spectrometer with Bruker accessory (EQUINOX 5). Spectra were automatically corrected for water, carbon dioxide and the KBr background (Monarumit et al., 2015; Zhang et al., 2016; Cardoso et al., 2016).

Chemical analyses were also done using the semi-quantitative method of X-ray fluorescence spectrometry (XFR) with wave dispersion in a spectrometer (Bruker S4 Pioneer) with a 4kW excitation source. A vacuum scan was done of 71 elements (11Na-92U) using an RX tube with Rh anode, 25–60 kV excitation voltage, and a 0.46 dg collimator with a 34 mm mask. Data was interpreted with the Spectra Plus software. Quantification of CaCO3 was done with an ignition loss analysis at 950 °C for 1 h (Shi et al., 2018).

The CaCO3 crystalline phase was identified using a Raman spectrometer with focal point (WITEC Alpha 300) (λexc = 488 and 785 nm; acquisition time 10 s; resolution 10 cm−1).

Results

Microstructure and nanostructure

The shell of P. colymbus consists of a prismatic and an aragonitic layer clearly divided by a slight change in color (Fig. 1, indicated by arrow and BL symbol). The FESEM analysis identified an organic hydrogel on some tablets in different layers (Fig. 2A). In another growth section of the pearlescent layer small growing crystals were observed to be fusing with each other (Fig. 2B). The tablets form a uniform sheet at the point where the next layer begins to grow and in some cases the tablets fuse with higher layers (Fig. 2C). Mature tablets are four to eight-sided, 0.6–3.0 µm in length, and can be fused (Fig. 2D).

Figure 1 Scanning electron microscopy (SEM) image of the cross-section of Pteria colymbus (Mollusca, Bivalvia) shell.

General view of pearl oyster Pteria colymbus (Mollusca, Bivalvia) shell showing the prismatic layer (PL) at the lower end, followed by the interface zone (BL) and the nacreous layer (NL) at the top. Middle zone of the nacreous layer (IL).

Figure 2 Scanning electron microscopy (SEM) view of the inner surface of the Pteria colymbus (Mollusca, Bivalvia) shell.

Microstructure of the nacreous tablets in the inner surface of the Pteria colymbus shell. The sample did not receive any polishing and etching treatment. (A) Detail of the inner surface of the shell showing recently formed nacre tablets and traces of the hydrogel (arrow) responsible for the formation of the nacre tablets. (B) Detail showing the first stages of nacre crystal formation. Crystal enveloped by organic matrix (★). Crystal in an advanced stage of formation merging with neighboring crystals (). (C) Section showing the two different layers of nacre growth merging through a hexagonal-shaped tablet (arrow). (D) Detail of the last layer of nacre showing the different geometries of the nacre tablets. tablet with the minimum number of sides for this species with only 5 µm in length (★). Tablet fused with others with no apparent pattern, which is 40 µm long ().

Cross sections of the nacre tablets showed a nanostructure in which the first layer of deposited nacre consisted of packed nanocrystals forming tablets. These were longer than in the upper layers and their growth was perpendicular to the surface (Fig. 3A). Crystals in initial growth stages were observed in a section close to the prismatic layer (Fig. 3B). In the intermediate section of the shell all the deposited nacre layers contained tablets composed of a package of crystals fused in the “brick and mortar” arrangement typical of species in the Pteriidae family. Average tablet thickness was 385 nm (SD = 0.069) (n = 187) with a range of 200–530 nm (Fig. 3C). The nanocrystals were uniformly oriented perpendicular to shell surface, have a 41 nm (SD = 9.43) (n = 24) average width, and a length range of 24–69 nm (n = 24) (Fig. 3D).

Figure 3 Scanning electron microscope view of the transverse section of Pteria colymbus (Mollusca, Bivalvia) shell.

Microstructure of the first and middle nacre layers in the shell of Pteria colymbus. (A) Interface zone of the prismatic and nacreous layers. The first layer of nacre crystals has a growth direction perpendicular to the surface (Arrow). The crystals have a greater length compared to the subsequent layers. The next three nacre layers (NL) have better organization and uniform size. (B) Detail of the nacre crystals (NL) in the first stage of growth. The presence of organic hydrogel (H) is observed, which is covering the prismatic layer (PL). (C) Middle zone of the nacre layer. All nacreous layers contain tablets composed of fused individual crystals. The typical brick and mortar formation, typical of bivalve mollusks (represented by white lines) is observed. The average thickness of the tablets was 385 nm (SD = 0.069). (D) Detail of the nacreous layers in the middle zone. The uniform shapes of the tablets are observed, the crystals that make up the tablets show an early stage of growth (★). Crystals merging from the center in the previous layer (). the crystals have a mean width of 41 nm (SD = 9.43).

Chemical analysis

The XRD pattern of the nacre layer showed the main reflection to be at 31.05 (2θ) with two lesser readings at 33.0 (2θ) and 66.0 (2θ), both characteristic of aragonite (Fig. 4).

Figure 4 X-ray diffraction pattern (XRD) of Pteria colymbus (Mollusca, Bivalvia) nacre sample.

The stronger diffraction peak of aragonite in nacreous layer of Pteria colymbus is scanning angle 2θ = 31.05°. Red bars represent the Aragonite pattern from PDF 041-1475. (For interpretation of the references to color in this figure legend, the reader is referred to the web version of this article).

The XRF analysis identified aragonite as representing 98.13% of sample weight. Minor elements were Si (0.72%), Na (0.5500%), S (0.2080%), and Sr (0.1042%), and trace elements were Mg (0.0897%), P (0.0485%), Cu (0.0353%), Al (0.0331), Fe (0.0311%), Cl (0.0290%), K (0.0151%) and Zn (0.0060%). The nacre’s aragonite structure was confirmed in the FTIR analysis (Fig. 5). Four bands characteristic of aragonite, corresponding to the CO32− ions, were identified: v3 at 1,445.96 cm−1; v1 at 1,082.72 cm−1; v2 at 856.80 cm−1; and v4 at 699.81–712.52 cm−1. The v4 band corresponds to the planar flexion mode of carbonate vibration and the v1 band to the symmetric stretch mode. The nacre FTIR spectrum revealed lower intensity organic bands; the band at 1,786.95 cm−1 was attributed to the carboxylate (carbonyl) groups of the acidic proteins in the organic matrix.

Figure 5 Fourier transform infrared spectrometry (FTIR) spectrum of Pteria colymbus (Mollusca, Bivalvia) nacre powder.

The nacre powder sample of Pteria colymbus displays a characteristic symmetric carbonate stretching vibration (v1) at 1,082.72 cm−1 and a carbonate out-of-plane bending vibration (v2) at 856.80 cm−1. The v4 band at 699.81–712.52 cm−1 corresponds to the planar flexion mode of carbonate vibration.

The Raman spectroscopy analysis of the nacre surface produced the most intense band at near 1,085 cm−1 in the aragonite spectra which corresponds to symmetrical stretching mode v1 of the carbonate ion (Fig. 6). Low- to medium-intensity bands in the 100–300 cm−1 region of the aragonite spectra were due to the translational and rotational modes of lattice vibration. The v4 in the carbonate ion plane bending mode occurred as a doublet with bands between 701 and 705 cm−1.

Figure 6 Raman spectra of the Pteria colymbus (Mollusca, Bivalvia) nacre layer sample.

The nacreous layer of Pteria colymbus shows an intense band at near 1,085 cm−1 that corresponds to the symmetrical stretching mode of the carbonate ion. Low- to medium-intensity bands in the 100–300 cm−1 region were due to the translational and rotational modes of lattice vibration. The carbonate ion plane bending mode occurred as a band between 701 and 705 cm−1.

Discussion

Nacre in P. colymbus consists of polygonal tablets with four to eight sides, a morphological characteristic also present in species such as Pinctada maxima, Pinctada radiata and Pinctada fucata (Wang et al., 2001; Bellaaj-Zouari et al., 2011; Zhang, Xie & Yan, 2019). However, tablet length differs between species in the Pteriidae family. In P. colymbus, length ranges from 0.6 to 3.0 µm, whereas in P. radiata it ranges from 4.0 to 5.0 µm (Bellaaj-Zouari et al., 2011) and in P. margaritifera from 5.0 to 10.0 µm (Rousseau et al., 2009).

Morphological differences in pearl oysters can be seasonal or intrinsic to each species (Wada, 1972). Intrinsic differences are due to genetic control of shell mineralogy and any similarities between species can be traced to the Mesozoic/Paleozoic boundary (Kennedy, Taylor & Hall, 1969). Environmental factors such as temperature can modify shell structure in different organisms. For instance, shell aragonite percentage in the mussel Mytilus californianus decreases from 45% in the spring to 30% in the winter, but only in organisms longer than 15 mm (Kennedy, Taylor & Hall, 1969). Slower nacre deposition rates may be in response to reductions in water temperature, probably in winter, combined with lower food availability (Taylor & Strack, 2008). In another example, in P. fucata nacre growth and thickness respond to tablet thickness which is 324 nm when water temperature is highest, during August, but only 224 nm when it is lower, in December (Muhammad et al., 2017). Decreases in tablet thickness favor light iridescence of the nacre surface which is why pearl producers harvest pearls when sea water temperature decreases (Suzuki & Nagasawa, 2013).

The tablets in P. colymbus shell contain nanocrystals ranging in thickness from 24 to 69 nm. This coincides with tablets in P. maxima which consist of aragonite nanofibrils from 10 to 30 nm thick (Wang et al., 2013). Tablets in abalone Haliotis rufescens shell are built of parallel aragonite nanoparticles (Huang & Li, 2012), and its nanostructure is one of polygonal cobble-like grains (~32 nm) within individual aragonite tablets (Li et al., 2004). The shell of P. fucata contains nanoblocks from 20 to 180 nm long (Oaki & Imai, 2005).

In each species formation of the aragonite nanoparticles in nacre tablets is regulated by organic matrix characteristics (Dauphin & Denis, 2000; Kim et al., 2006). The distribution of organic macromolecules during crystal growth is important since these regulate crystal size, shape, and orientation (Okumura et al., 2012; Shtukenberg, Ward & Kahr, 2017).

In Tellinella asperrima shells, the crystallographic orientation of aragonite between nacre and nacre powder (ground at 45 µm) was investigated. It was found that there is a strong preferential crystallographic orientation of the diffraction peak at the angle 2θ = 31.03° of the intact nacre, simultaneously two other peaks with weak preferential orientations were found (Ren et al., 2009). In this work, it was found that the X-ray diffraction pattern of the nacre of P. colymbus is very similar to the intact nacre of T. asperrima. Presenting greater intensity in the main diffraction peak and the same intensity in the minor peaks. This preferential crystal orientation of aragonite in nacre is frequently found in biological mineralization and is due to the presence of b-chitin fibrils and the protein polypeptide chains that control crystal growth (Feng et al., 2000).

Calcium carbonate (CaCO3) in the form of aragonite represents 98.13% of the total nacre layer in P. colymbus in the present results. The majority presence of aragonite in the shell coincides with other species in the Pteriidae family, such as P. fucata (Saruwatari et al., 2009) and P. margaritifera (Shi et al., 2018), as well as the mussel Perna viridis (Xu & Zhang, 2015). Minor elements are Si, Na, S and Sr, and trace elements are Mg, P, Cu, Al, Fe, Cl, K and Zn. In another study, nacre was found to be composed of calcium carbonate (91.50%) with traces of organic substances (3.83%), residual substances (0.01%) and water (3.97%) (Taylor & Strack, 2008). These residual substances include Na, Cl and K, as well as traces and other elements such as Ba, Mg, P, Mn, Fe, Al, Cu, Zn, Ag, Hg, Li and Mr. The trace element profile of a nacre reflects water mineral composition in the place where it formed. In biogenic aragonite crystals other elements such as Mn, Mg, Sr and Ba can substitute for calcium (Gaetani & Cohen, 2006; Chen et al., 2011). Notable differences in the concentration of these elements exist between taxonomic groups, highlighting genetic influence on their incorporation (Carré et al., 2006).

The other elements present in nacre function as precursor ions in nacre formation, as catalysts in proteins and activators of enzymes; they are present in intercrystalline organic macromolecules and the organism’s epithelial fluid (Marsh & Sass, 1983; Cho & Jeong, 2011; Marin, 2012).

Conclusions

Nacre composition in P. colymbus has high aragonite content and its nanostructure consists of polygonal tablets built of nanocrystals. The present is a preliminary description of P. colymbus shell structure intended as a presentation of the relevant data to date, and a guide for further research.

Supplemental Information

Supplemental Information 1 XRD, FTIR and Raman data.

Click here for additional data file.

The authors thank Patricia Quintana access to LANNBIO facilities, Dora Huerta, and Daniel Aguilar for assistance with the diffractograms and SEM images. The authors also thank Maria del Socorro Garcia Guillermo, Ixchel Rubí Perez, Ana Elena Muñiz, and Norma Alicia Berlanga, at Cinvestav-Saltillo chemical analysis General Laboratory for her support in carrying out this work.

Additional Information and Declarations

Competing Interests

Author Contributions

Data Availability

The authors declare that they have no competing interests.

Pablo Santana conceived and designed the experiments, performed the experiments, analyzed the data, prepared figures and/or tables, authored or reviewed drafts of the paper, and approved the final draft.

Dalila Aldana Aranda conceived and designed the experiments, performed the experiments, analyzed the data, prepared figures and/or tables, authored or reviewed drafts of the paper, and approved the final draft.

The following information was supplied regarding data availability:

Raw measurements are available in Supplemental File.

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
