# Peer review of "Nacre morphology and chemical composition in Atlantic winged oyster Pteria colymbus (Röding, 1798)"

_PeerJ, doi:10.7717/peerj.11527_

## Round 0.1 · original submission · Minor Revisions

Dear Dr Santana,

Please receive the referees' comments on your article.

They recommend publication after minor revision.

Please consider all the point sraised and provide a revised version of your article

Reviewer 1 ·

Basic reporting

The article is concise and clearly written in a sufficiently correct English.
The Reference format is not that requested by the Journal.
The figures are well done but there are no captions to the figures.

Experimental design

The research topic is within the scope of the journal.
The methods are described with sufficient detail and the investigation conducted in a rigorous manner.

Validity of the findings

no comment

Additional comments

Just a few specific comments:
- the technique used is the powder XRD, so I would add the word powder at line 79 and throughout the text.
- At line 92 please correct Briker with Bruker.
- Please rephrase the sentence at lines 152-155.

Reviewer 2 ·

Basic reporting

The paper by Santana and Aranda reports a preliminary characterization of the structure of Pteria colymbus oyster shell by using an appropriate set of experimental techniques. The results are sound and, in my opinion, this work can be published after only minor revisions as outlined below.

Experimental design

In the Materials & Methods section, the Authors say that six 1 cm2 samples were cut from the shells. For sake of reproducibility of the experiments, a sentence on how powder samples for XRD and FTIR measurements have been prepared should be added.
Figure 4 reports the XRD pattern of nacre powder. However, the spectrum presents few reflections and it resembles the spectrum of the nacre from mussels, while it differs from the XRD spectrum of powdered mussel shells, as reported by Ren et al. (see reference list). The Authors should add a comment about this (i.e.: does it depends from a different grinding procedure?). Figure 4 should also report the position and indexing of aragonite reflections present in the reference ICDD card (no. 76-606) for comparison.
The title of the X-axis of Figure 4 is better written as 2θ.
The title of the X-axis of Figure 5 is wrong. It should report wavenumbers in cm-1. Also, the Y-axis of this figure should read as just “Intensity” or “Intensity / a.u.”

Validity of the findings

As stated above, the findings of this study are valuable and should be published.

---

## Round 0.2 · accepted · Accept

In the revised version all the referees' requests have been fulfilled so the paper can be accepted for publication.

---

## Author Rebuttal · Round 0.2

Cinvestav
UNIDAD MERIDA

April 30, 2021

Dear Andrea Scozzafava,

Academic Editor, PeerJ.

Please find attached comments and corrections to article no. 57287 entitled "Nacre morphology and chemical composition in Atlantic winged oyster *Pteria colymbus* (Röding, 1798)" by the authors: Dalila Aldana Aranda and Pablo Santana Flores.

We thank the reviewers for their generous comments on the manuscript. We have incorporated all comments.

Best regards,

Dr. Dalila Aldana Aranda
Profesor
Cinvestav IPN
daldana@cinvestav.mx

# Reviewer 1

## Basic reporting

*1.- The Reference format is not that requested by the Journal.*

The reference format was corrected according requested by the journal.

*2- The figures are well done but there are no captions to the figures.*

The captions to the figures were corrected as follows:
Fig. 1 Title: Scanning electron microscopy (SEM) image of the cross-section of *Pteria colymbus* (Mollusca, Bivalvia) shell.

Legend: General view of pearl oyster *Pteria colymbus* (Mollusca, Bivalvia) shell showing the prismatic layer (PL) at the lower end, followed by the interface zone (BL) and the nacreous layer (NL) at the top. Middle zone of the pearly layer (IL).

Fig. 2. Title: Scanning electron microscopy (SEM) view of the inner surface of the *Pteria colymbus* (Mollusca, Bivalvia) shell.

Legend: Microstructure of the nacreous tablets in the inner surface of the *Pteria colymbus* shell. The sample did not receive any polishing and etching treatment. (a) detail of the inner surface of the shell showing recently formed nacre tablets and traces of the hydrogel (arrow) responsible for the formation of the nacre tablets. (b) Detail showing the first stages of nacre crystal formation. Crystal enveloped by organic matrix (★). Crystal in an advanced stage of formation merging with neighboring crystals (✹). (c) Section showing the two different layers of nacre growth merging through a hexagonal-shaped tablet (arrow). (d) Detail of the last layer of nacre showing the different geometries of the nacre tablets. tablet with the minimum number of sides for this species with only 5 µm in length (★). Tablet fused with others with no apparent pattern, which is 40 µm long (✹).

Fig. 3 Title: Scanning electron microscope view of the transverse section of *Pteria colymbus* (Mollusca, Bivalvia) shell.

Legend: Microstructure of the first and middle nacre layers in the shell of *Pteria colymbus*. (a) Interface zone of the prismatic and nacreous layers. The first layer of nacre crystals has a growth direction perpendicular to the surface (Arrow). The crystals have a greater length compared to the subsequent layers. The next three layers of mother-of-pearl (NL) have better organization and uniform size. (b) Detail of the nacre crystals (NL) in the first stage of growth. The presence of organic hydrogel (H) is observed, which is covering the prismatic layer (PL). (c) Middle zone of the nacre layer. All nacreous layers contain tablets composed of fused individual crystals. The typical brick and mortar formation, typical of bivalve mollusks (represented by white lines) is observed. The average thickness of the tablets was 385 nm (SD = 0.069). (d) Detail of the nacreous layers in the middle zone. The uniform shapes of the tablets are observed, the crystals that make up the tablets show an early stage of growth (★). Crystals merging from the center in the previous layer (✹). the crystals have a mean width of 41 nm (SD = 9.43).

Fig. 4. Title: X-ray diffraction pattern (XRD) of nacre sample of *Pteria colymbus* (Mollusca, Bivalvia) shell.

Legend: The stronger diffraction peak of aragonite in nacreous layer of *Pteria colymbus* is scanning angle 2θ = 31.05°.  Red bars represent the Aragonite pattern from PDF 041-1475. (For interpretation of the references to color in this figure legend, the reader is referred to the web version of this article).

Fig.5. Title: FTIR spectrum of *Pteria colymbus* (Mollusca, Bivalvia) nacre powder.
Legend: The nacre powder sample of *Pteria colymbus* displays a characteristic symmetric carbonate stretching vibration (v1) at 1082.72 $cm^{-1}$ and a carbonate out-of-plane bending vibration (v2) at 856.80 $cm^{-1}$. The v4 band at 699.81-712.52 $cm^{-1}$ corresponds to the planar flexion mode of carbonate vibration.

Fig.6 Title: Raman spectra of the *Pteria colymbus* (Mollusca, Bivalvia) nacre layer sample.

Legend: The nacreous layer of *Pteria colymbus* shows an intense band at near 1085 $cm^{-1}$ that corresponds to the symmetrical stretching mode of the carbonate ion. Low- to medium-intensity bands in the 100-300 $cm^{-1}$ region were due to the translational and rotational modes of lattice vibration. The carbonate ion plane bending mode occurred as a band between 701 and 705 $cm^{-1}$.

*3.- the technique used is the powder XRD, so I would add the word powder at line 79 and*

*throughout the text.*

We add the word nacre sample, in the X-ray analysis. We do not use powder samples.

*4.- At line 92 please correct Briker with Bruker.*

The word Bruker was corrected.

*5.- Please rephrase the sentence at lines 152-155.*

The sentence on lines 152-155 was rewritten as follows:

In another example, in P. fucata the growth and thickness of the nacre respond to the thickness of the tablet, which is 324 nm when the water temperature is highest, during August, but only 224 nm when it is lowest, in December (Muhammad et al. 2017).

## Reviewer 2

**Experimental design**

*In the Materials & Methods section, the Authors say that six 1 cm2 samples were cut from the shells. For sake of reproducibility of the experiments, a sentence on how powder samples for XRD and FTIR measurements have been prepared should be added.*

We add a paragraph mentioning the preparation of the mother of pearl sample for the XRD analysis in lines 78-82. We also added a paragraph mentioning how nacre powder was prepared for FTIR analysis on lines 87-91.

*Figure 4 reports the XRD pattern of nacre powder. However, the spectrum presents few reflections and it resembles the spectrum of the nacre from mussels, while it differs from the XRD spectrum of powdered mussel shells, as reported by Ren et al. (see reference list). The Authors should add a comment about this (i.e.: does it depends from a different grinding procedure?). Figure 4 should also report the position and indexing of aragonite reflections present in the reference ICDD card (no. 76-606) for comparison.*

We added the reference diffraction pattern of aragonite in figure 4. We added a comment about the similarity between the diffraction patterns of the unground sample of P. colymbus and the unground sample of *T. asperrima*, this paragraph is found at lines 173-181, as follows:

In *Tellinella asperrima* shells, the crystallographic orientation of aragonite between nacre and nacre powder (ground at 45 μm) was investigated. It was found that there is a strong preferential crystallographic orientation of the diffraction peak at the angle $2\Theta = 31.03$ ° of the intact nacre, simultaneously two other peaks with weak preferential orientations were found (Ren et al. 2000). In this work, it was found that the X-ray diffraction pattern of the nacre of *P. colymbus* is very similar to the intact nacre of *T. asperrima*. Presenting greater intensity in the main diffraction peak and the same intensity in the minor peaks. This preferential crystal orientation of aragonite in mother-of-pearl is frequently found in biological mineralization and is due to the presence of b-chitin fibrils and the protein polypeptide chains that control crystal growth. (Feng et al 2000).

*The title of the X-axis of Figure 4 is better written as 2θ.*

We corrected the title of the X-axis of figure 4.

*The title of the X-axis of Figure 5 is wrong. It should report wavenumbers in cm-1. Also, the Y-axis of this figure should read as just "Intensity" or "Intensity / a.u."*

The title of the X-axis and Y-axis was corrected according to suggest of reviewers.